# A Joint Method Based on Geochemistry and Magnetotelluric Sounding for Exploring Geothermal Resources in Sedimentary Basins and Its Application

Yanguang Liu [1,2], Guiling Wang [1,2,*], Xuezhong Guo [3], Jing Hu [4,*], Jianguo Wang [5,*], Xiaojun Wang [6] and Gui Zhao [2]

1   Institute of Hydrogeology and Environmental Geology, Chinese Academy of Geological Sciences, Shijiazhuang 050061, China
2   Technology Innovation Center for Geothermal & Hot Dry Rock Exploration and Development, Ministry of Natural Resources, Shijiazhuang 050061, China
3   Western Mining Co., Ltd., Xining 810016, China
4   School of Civil Engineering and Architecture, Jiangsu University of Science and Technology, Zhenjiang 212003, China
5   Department of Geological Engineering, Qinghai University, Xining 810001, China
6   Hydrological Geological Team of Hebei Province Coal Geology Bureau, Handan 056000, China
*   Correspondence: ihegwangguiling@sina.com (G.W.); 15736315432@139.com (J.H.); lywjg467047@126.com (J.W.)





Abstract: The precise exploration of the characteristics of geothermal fields in sedimentary basins, such as the temperature and burial depth of their deep geothermal reservoirs, is of great significance for improving the probability of penetration and reducing exploration risks and development costs. This study proposed a joint exploration method combining magnetotelluric (MT) sounding and geothermometers. Using this method, this study estimated the geothermal reservoirs' temperature and the circulation depth of geothermal water in the Xianxian geothermal field, a typical geothermal field in a large sedimentary basin in northern China, and prepared the temperature and depth maps of the geothermal reservoirs. The main results are as follows. First, the bedrock's geothermal reservoirs with karst fissures in the Xianxian geothermal field have great potential for development. Among them, geothermal reservoirs in the Jixianian Wumishan formation have a top depth of 1100–1500 m and a thickness of 700–1700 m, and the geothermal reservoirs in the Jixianian Gaoyuzhuang formation have a top depth of 3700–4000 m and a maximum drilled thickness of 400 m. The geothermal reservoirs of the Xianxian geothermal field mainly have medium and low temperatures of 138–160 °C and the circulation depth of the geothermal water is 5873 m.

Keywords: geothermal reservoirs in sedimentary basins; magnetotelluric (MT) sounding; fluid geothermometer; geothermal reservoir exploration

## 1. Introduction

In the context of the accelerated development of a low-carbon and circular economy, the exploration and development of geothermal resources have boomed worldwide. Geothermal resources are mainly divided into those in uplifted mountains and those in sedimentary basins, according to the regional structure and the reservoir's characteristics [1–3]. Sedimentary basins, especially large ones, have favorable conditions for the occurrence of geothermal resources. The reserves of medium- and low-temperature geothermal resources account for more than 90% of the total geothermal resources in China. Most medium- and low-temperature geothermal resources are contained in Mesozoic–Cenozoic sedimentary basins and have become the key targets for exploration and development [4]. The exploration of geothermal resources is the basis for their development and utilization and is also

an important field of application for geophysical and geochemical exploration methodologies [5,6]. Attempts have been made to utilize gravity, magnetic, electrical, seismic, and log surveys for geothermal exploration [7,8]. For example, resistivity anomalies converted from magnetic anomalies can be used to determine the locations of deep-seated faults and stratigraphic features. Electrical properties can be used to characterize shallow reservoirs, delineate geothermal reservoirs, and clarify deep thermal mechanisms and processes of thermal evolution [9]. Among the various geophysical methods, MT sounding is the most widely used method in geothermal exploration, with good performance and high precision. Geochemical methods are also commonly used in geothermal exploration. Among them, fluid geochemistry can be used to extract information on the deep thermal state. Moreover, it performs well in predicting the areas of occurrence of geothermal fields, estimating the temperature of deep geothermal reservoirs, and inferring the genesis of geothermal waters [10–12]. As a result, this method has become a necessary means for exploring hydrothermal fields. Each geophysical and geochemical exploration methodology has its advantages and characteristics. Therefore, it is necessary to construct a cost-effective combined methodology that can accurately identify the geothermal reservoir's structure in geothermal exploration.

Temperature is one of the key features of the Earth's interior, and knowledge about it determines the ability of humankind to conduct basic geoscientific research and geothermal applications. Therefore, it is critical to estimate the distribution of the subsurface temperatures accurately. Currently, the Earth's interior temperature is usually estimated on the basis of boreholes' temperature log data. However, spatial interpolation based on temperature log curves measured in several irregularly distributed wells often leads to considerable estimation errors. In the actual exploration of geothermal reservoirs, geophysical exploration methods are usually combined with geothermometers to predict geothermal reservoirs' temperature, thus improving the accuracy of exploration [13–16]. The temperature distribution of geothermal reservoirs can be intuitively reflected when geophysical methods such as MT are used to determine the depth and thickness of rock formations [17,18]. Some scholars have studied and analyzed the sensitivities of the formations' salinity and the formations' resistivity, and have indicated that the resistivity of formations with low groundwater salinity is significantly affected by changes in temperature [19]. According to the statistics of Yinsheng et al, the groundwater in most strata, especially in sedimentary strata, generally has a low total dissolved solid (TDS) content, which is within the range where the formation's resistivity is sensitive to changes in temperature, and the first-order change in temperature has a significant impact on the formation's resistivity [20]. On the basis of this finding, they developed an inversion method of measuring a geothermal reservoir's temperature based on resistivity derived by using the controlled source audio-frequency magnetotelluric (CSAMT) technique (also referred to as the CSAMT resistivity–temperature method).

The inversion of deep geothermal reservoirs' temperatures using geochemical methods (e.g., geothermometers) can be used to supplement and verify the results of geophysical exploration [21–23]. Geothermometers generally include cation and $SiO_2$ geothermometers and integrated multicomponent solute geothermometry, which are widely used in the estimation of geothermal reservoirs' temperatures [24–33]. Barcelona et al. [33] estimated the temperature of the Gollette geothermal field using a cation geothermometer and the results obtained were consistent with those achieved using multiple mineral equilibrium diagrams. Xu et al. [34] believed that the $SiO_2$ geothermometers were more suitable for estimating reservoirs' temperature than the silicon-enthalpy mixing model, and used them to estimate the temperature of the geothermal reservoir of the Xi'an geothermal field. In 2022, Zhao et al. [9] inverted the distribution characteristics of the temperature of the geothermal field of the formation in uplifted mountains using the CSAMT resistivity–temperature inversion method, and the results obtained were roughly consistent with the temperature measured in boreholes and the estimates of a chalcedony geothermometer, indicating that this method is feasible. However, geothermal reservoirs have complex

geological conditions, and the geothermal fields in sedimentary basins have significantly different stratigraphic structures and geothermal reservoir characteristics from those in uplifted mountains. Therefore, it is necessary to further research the resistivity–temperature inversion method to improve its applicability and enrich the theories it involves.

The upper part of a sedimentary basin includes high-porosity and high-permeability reservoirs composed of large quantities of coarse clastic materials and relative aquicludes comprising an appreciable amount of fine-grained materials, both of which are characterized by multi-cyclic and multi-layered sedimentation. The lower part of a basin shows the wide presence of Paleozoic, Mesoproterozoic, and Neoproterozoic carbonate sedimentary suites, creating effective deep geothermal reservoirs. Moreover, deep-seated faults are often present in a sedimentary basin. Together with the conductive heat driven by the terrestrial heat flow, they create the conditions for various and extensive geothermal reservoirs and a complex temperature distribution. The unclear distribution of the deep temperature field in a sedimentary basin leads to expensive and high-risk exploratory drilling. Therefore, there is an urgent need to determine economical and efficient technologies for the surface exploration of deep temperature fields. In line with previous studies, this study applied the resistivity–temperature method to the Xianxian geothermal field. First, this study analyzed the relationship between the MT resistivity and temperature of the strata based on the temperature measured in boreholes, and estimated the depth and temperature of the geothermal reservoirs in the geothermal field. Subsequently, it estimated the geothermal reservoir's temperature and the circulation depth of geothermal water using geochemical methods. This study laid a foundation for improving the precision of explorations of geothermal reservoirs.

## 2. Geological Setting

The Xianxian geothermal field spans two third-order tectonic units, i.e., the Cangxian uplift and the Jizhong depression (Figure 1), and consists of four fourth-order tectonic units, namely the Raoyang sag, the Xianxian uplift, the Fucheng sag, and the Qingxian uplift from west to east. As the most important fault in the study area, the Xianxian fault has a long extension, a large fault throw, a strike of NE–NNE, and a NW dip direction. It is the boundary fault between the subsiding Jizhong depression and the Cangxian uplift, resulting in the absence of the Guantao formation and the Lower Neogene deposits in the Xianxian area [4,35–38].

The statistics of the resistivity of formations in northern China (Table 1) show that the strata in the study area have the following common features: (1) the Mesoproterozoic strata have the highest resistivity, followed by the Lower Paleozoic strata and then the Mesozoic–Cenozoic strata; (2) the Mesoproterozoic Jixianian Wumishan formation has significantly different resistivity from the overlying Cenozoic strata; (3) the log curve data of the surrounding boreholes show that the Cenozoic Quaternary and Neogene strata in the study area are mainly composed of medium- and fine-grained sands, clays, loams, sandy loams, and sandy gravels, and have a resistivity less than 200 $\Omega \cdot m$ generally; and (4) the upper part of the Mesoproterozoic Jixianian strata are composed of clastic rocks, while the lower portion is dominated by gray and grayish-green dolomites, and these strata have a resistivity of over 800 $\Omega \cdot m$ [39].

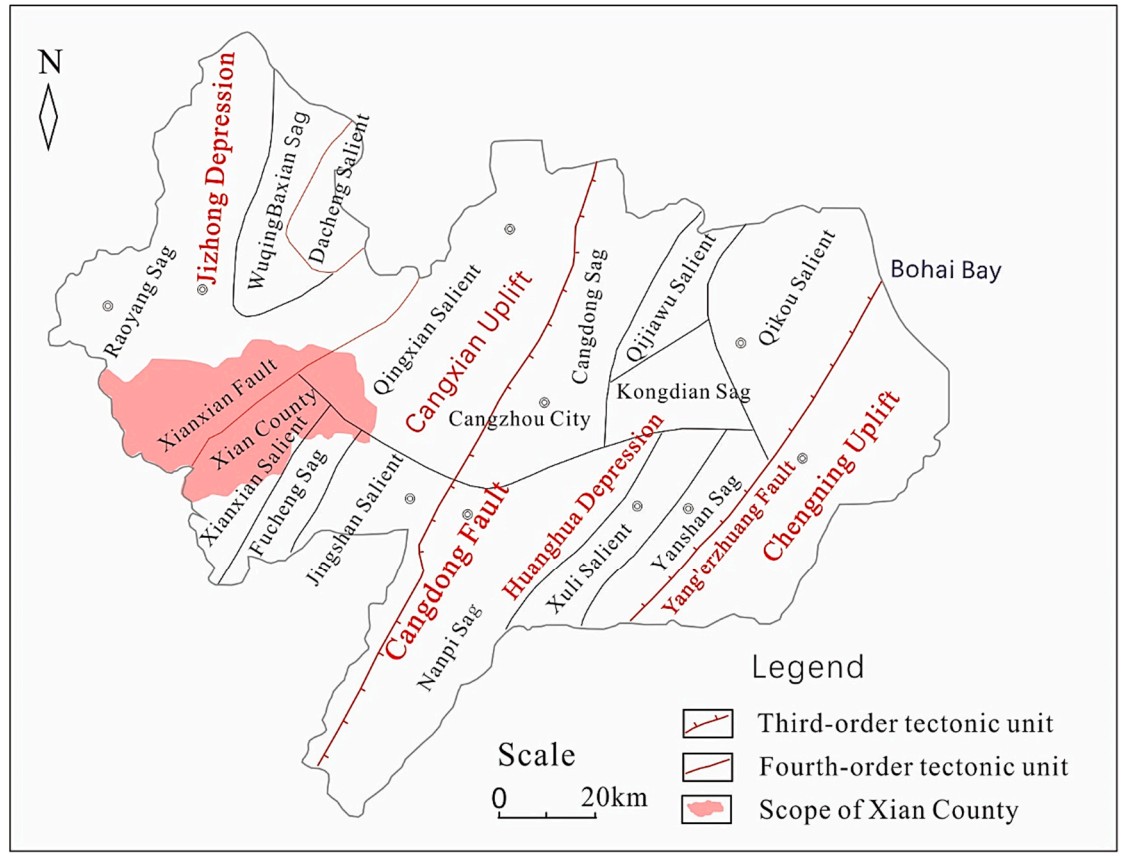

**Figure 1.** Sketch of the regional structures in Xianxian County.

**Table 1.** Statistics of the resistivity of formations in northern China.

| Erathem | System | Series | Formation | Member | Code | Resistivity (Ω·m) | Average Resistivity (Ω·m) |
|---|---|---|---|---|---|---|---|
| Cenozoic | Quaternary | Holocene and Pleistocene | | | Q | 11.0–29.0 | 17.5 |
| | Neogene | Pliocene | Minghuazhen | Upper member | $Nm_{upper}$ | 5.87–17.4 | 10.6 |
| | | | | Lower member | $Nm_{lower}$ | 4.35–8.65 | 6.0 |
| | | Miocene | Guantao | | Ng | 3.49–6.20 | 4.46 |
| | Paleogene | Oligocene | Dongying, Shahejie | | Ed + Es | 2.48–5.50 | 4.12 |
| | | Eocene | Kongdian | | $E_k$ | 3.60–12.9 | 6.54 |
| Mesozoic | Cretaceous, Jurassic, and Triassic | | | | Mz | 6.27–61.6 | 22.2 |
| Paleozoic | Permian and Carboniferous | | | | P-C | 17.7–37.2 | 22.7 |
| | Ordovician | | | | O | 120–470 | 209 |
| | Cambrian | | | | ∈ | 33.2–112.5 | 69.5 |
| Middle–Upper Proterozoic | Jixianian and Changchengian | | | | $Pt_{2–3}$ | 150–1162 | 517 |
| Archaeozoic | | | | | Ar | 2.8–204,184 | 1398.92 |

The significant differences in resistivity between the Jixianian dolomites and the Quaternary and Neogene fine sands and clays are favorable for exploring faults and determining strata, thus providing a basis for the exploration of geothermal resources.

## 3. Methods

### 3.1. MT Sounding

Three MT survey lines (L1, L2, and L3) were arranged in the study area, with a total length of 30,000 m and 35 survey points in total (the spacing between two points was 1000 m). Moreover, survey points were additionally deployed at the locations of known boreholes to enhance the reliability of the geological interpretation. The MT sounding was carried out using an MTU-5 geophysical instrument and a V8 receiver, both of which were produced by the Canadian company Phoenix. The MTU-5 operated at a frequency range of 0.00005–400 Hz and could be synchronized to high-precision GPS. The V8 operated at a frequency range of 0.00005–10,000 Hz and used an advanced modular design. They were compatible with each other and could achieve multi-station large-area synchronous acquisition, allowing the execution of 3D or 4D electrical surveys.

Processing of the magnetotelluric data was mainly conducted using Bostick 1D inversion and RRI 2D inversion in this study. Bostick 1D inversion is widely used in magnetic surveys, since it does not require an initial model and multiple iterations, and has a high computational speed. RRI 2D inversion is based on a stable and advanced algorithm, has a unique processing method for static effects, and provides a forward modeling section. Therefore, it was convenient to verify the inversion results using RRI 2D inversion. The data processing workflow was as follows. First, the raw MT data were preprocessed through denoising, time windowing, and spectral analysis to improve the signal-to-noise ratio (SNR) of the data; next, the necessary analyses of the parameter points, curve types, and phases were made to enhance the data's reliability; finally, the inverted resistivity sections were obtained using methods such as well point correction, Bostick 1D inversion, and RRI 2D inversion and were then compared with the known data to improve the geological interpretation of the results.

### 3.2. Collection and Analysis of Water Samples

Two sets of geothermal water samples were collected from the wells XXZK1 and XXZK2 and appropriately numbered (Figure 2). All geothermal water samples were filtered through 0.45 μm membranes and stored in three 250 mL high-density polyethylene bottles, which were rinsed twice using the water to be sampled before sample collection. For the $SiO_2$ analyses, the geothermal water samples were diluted to 10% of their initial concentration using deionized water. For the metallic and cation element analyses, the samples were acidified to pH 1 using $HNO_3$. No reagent was added to the samples for the analysis of inorganic anions. The orifice temperature, pH, Eh, and conductivity were measured using a portable waterproof pH/EC/TDS meter (HI991301).

The total chemical analysis and trace element tests were mainly conducted at the Key Laboratory of Groundwater Science and Engineering of the Department of Land and Resources, Institute of Hydrogeology and Environmental Geology, Chinese Academy of Geological Sciences. The cations' concentrations were detected using an Agilent 7700X ICP-MS at an ambient temperature of 26 °C and a humidity of 48%. The anions were analyzed using a VISTA-MPX plasma spectrometer at an ambient temperature of 20 °C and a humidity of 50%. The detailed methodology referred to the DZ/T0064-1993 Groundwater Test Methods and the GB/T8538-2008 Methods for Examination of Drinking Natural Mineral Water.

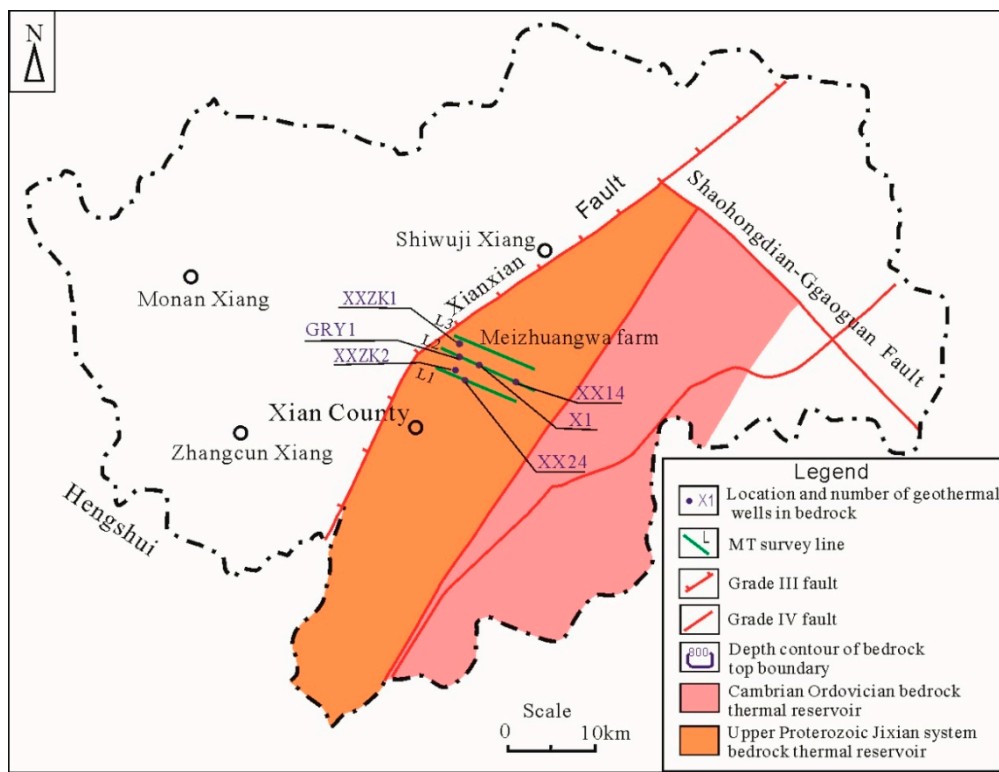

**Figure 2.** Distribution of the geothermal reservoirs of the bedrock and the locations of wells used for tracer tests in Xianxian County.

### 3.3. Estimation of the Geothermal Reservoirs' Temperature

### 3.3.1. Water–Rock Equilibrium State

The triangular Na–K–Mg diagram is frequently used to distinguish between fully equilibrated, partially equilibrated, and immature geothermal waters [24]. It can be used to explain the state of water–rock equilibrium in deep geothermal fluids. It is established on the basis of the reaction of the solubility of sodium and potassium minerals to temperature, and the reaction equation is as follows:

$$K_2O \cdot Al_2O_3 \cdot 6SiO_2 + Na^+ \rightarrow NaAlSi_3O_8 + K \tag{1}$$

$$2.8K_2O \cdot Al_2O_3 \cdot 6SiO_2 + 1.6H_2O + Mg^{2+} \rightarrow 0.8KAl_2(AlSi_3O_{10})(OH)_2 + 0.2Chlorite + 5.4Si + 2K^+ \tag{2}$$

The coordinates of the water samples' points in the triangular Na–K–Mg diagram can be calculated using the following equations

$$S = C_{Na}/1000 + C_K/100 + \sqrt{C_{Mg}} \tag{3}$$

$$Na\% = C_{Na}/10S \tag{4}$$

$$K\% = C_K/S \tag{5}$$

$$Mg\% = 100\sqrt{C_{Mg}}/S \tag{6}$$

where $C_{Na}$, $C_K$, and $C_{Mg}$ are the mass concentrations of Na, K, and Mg ions in the water, respectively, in mg/L.

### 3.3.2. Resistivity–Temperature Method

As early as the 1970s, the relationship between a formation's resistivity and temperature was summarized through numerous experiments and can be expressed as follows

$$\rho_t = \rho_0 / [1 + \alpha(T - 20)] \tag{7}$$

where $\rho_0$ is the formation's resistivity at 20 °C, which can be regarded as the initial value; $\rho_t$ is the formation's resistivity at T °C and is a measured value; and $\alpha$ is the temperature coefficient of a rock and is usually 0.02.

Transforming Equation (7) yields the expression for the temperature of geothermal reservoirs:

$$T = 20 + (\rho_0 / \rho_t - 1) / \alpha \tag{8}$$

The inverted resistivity of the formation can be obtained according to the MT survey data, and the strata can be divided into multiple zones based on the drilling data. By setting the initial resistivity of each zone and the temperature coefficient of a rock, the geothermal reservoir's temperature can be calculated using Equation (8). In this study, the initial resistivity of each zone was obtained using the linear interpolation method. Moreover, $\rho_0$ was set to the resistivity of the strata at an average depth, and the $\rho_0$ at other depths was calculated using the linear interpolation method to ensure that $\rho_0$ gradually increased with depth, thus eliminating the effects of depth variations on the formation's resistivity.

### 3.3.3. Geothermometers

(1) Geochemical geothermometry

Geothermometers are used to evaluate the temperature of deep geothermal reservoirs based on the empirical equations for the contents of some chemical components in geothermal fluids and the temperature of deep geothermal reservoirs. Geothermometers were developed on the following principle. When some chemical components in deep geothermal fluids reach a dissolution equilibrium state with the surrounding rocks, the geothermal water cools down but the chemical composition remains almost unchanged during upwelling [40]. In this study, the temperature of deep geothermal reservoirs in the Xianxian geothermal field was jointly determined using $SiO_2$ geothermometers, a K–Mg geothermometer, and multiple mineral equilibrium diagrams.

The equation for quartz geothermometers with no steam loss [41] is as follows

$$T = \frac{1309}{5.19 - \log(c_{SiO_2})} - 273.15 \tag{9}$$

where $T$ is the estimated value of the geothermal reservoir's temperature in °C and $c_{SiO_2}$ is the mass concentration of $SiO_2$ in water, in mg/L.

The equation for chalcedony geothermometers [41] is as follows

$$T = \frac{1032}{4.69 - \log(c_{SiO_2})} - 273.15 \tag{10}$$

where $T$ is the estimated value of the geothermal reservoir's temperature (°C) and $c_{SiO_2}$ is the mass concentration of $SiO_2$ in water, in mg/L.

In addition, the equation for $SiO_2$ content is as follows

$$C_{SiO_2} = M_{SiO_2} \times C_{H_2SiO_3} / M_{H_2SiO_3} \tag{11}$$

where $C_{SiO_2}$ and $C_{H_2SiO_3}$ are the concentrations of $SiO_2$ and $H_2SiO_3$ in the geothermal water, respectively (mg/L); $M_{SiO_2}$ is the relative molecular mass of $SiO_2$, which is 60; and $M_{H_2SiO_3}$ is the relative molecular mass of $H_2SiO_3$, which is 78.

The K-Mg geothermometer, developed on the basis of the ion-exchange reactions in which potassium feldspar transitions into muscovite and clinochlore, responds rapidly to

changes in temperature, and can quickly reach equilibrium in a solution. Therefore, it was applied to low-temperature geothermal water systems.

The equation for the K-Mg geothermometer is as follows

$$T = \frac{4410}{14.0 - \lg(\frac{C_K^2}{C_{Mg}})} - 273.15 \tag{12}$$

where $C_{Mg}$ is the concentration of Mg ions in th water.

(2)  Multiple mineral equilibrium diagrams

Reed and Spycher proposed the multiple mineral equilibrium diagrams in 1984 to determine the chemical equilibrium state between fluids and minerals in a geothermal system. This method was developed on basis of the following principle. Under the assumption that the dissolution of multiple minerals in geothermal water is a function of temperature, it can be determined that the geothermal water and these minerals reach equilibrium when all these minerals approximate equilibrium at a certain temperature. The temperature corresponding to the equilibrium state is the deep geothermal reservoir's temperature.

The saturation index (SI) of minerals in geothermal water can be used to determine the degree of saturation (SR) of the minerals. SI > 0, SI = 0, and SI < 0 denote the supersaturated, saturated, and undersaturated states, respectively. The formula for SI is as follows

$$SI = \lg(IAP/K) \tag{13}$$

where $K$ and $IAP$ are the solubility and the ion activity product (IAP) of a mineral in the geothermal water, respectively (mol/L).

*3.4. Calculation of the Circulation Depth of Geothermal Water*

The methods commonly used for calculating the circulation depths of geothermal fluid include the method using the equation for the circulation depth of geothermal water (the geothermal water circulation equation method) and the isotope method. Since data on the temperature measurement curves of Xianxian County were collected in this study, the geothermal water circulation equation method was used to calculate the circulation depth of geothermal water. The equation for the circulation depth of geothermal water is as shown below

$$Z = G(T_Z - T_0) + Z_0 \tag{14}$$

where $Z$ is the circulation depth of geothermal water, in m; $G$ is the reciprocal of the geothermal gradient, in m/°C; $T_z$ is the geothermal reservoir's temperature, in °C; $T_0$ is the annual mean temperature of the study area, in °C; and $Z_0$ is the thickness of the constant-temperature zone, in m.

## 4. Results and Discussion

*4.1. Analysis of Sections Obtained from MT Inversion*

Three MT survey lines, namely L1, L2, and L3 (Figure 2), were deployed in parallel and fully covered the study area, where four geothermal wells, namely X1, XX14, XX24, and GRY1, are located. The integrated interpretation was carried out through 2D inversion of the resistivity and frequency, as well as known drilling data. The 2D integrated interpretations of sections of the survey lines are shown in Figures 3–5.

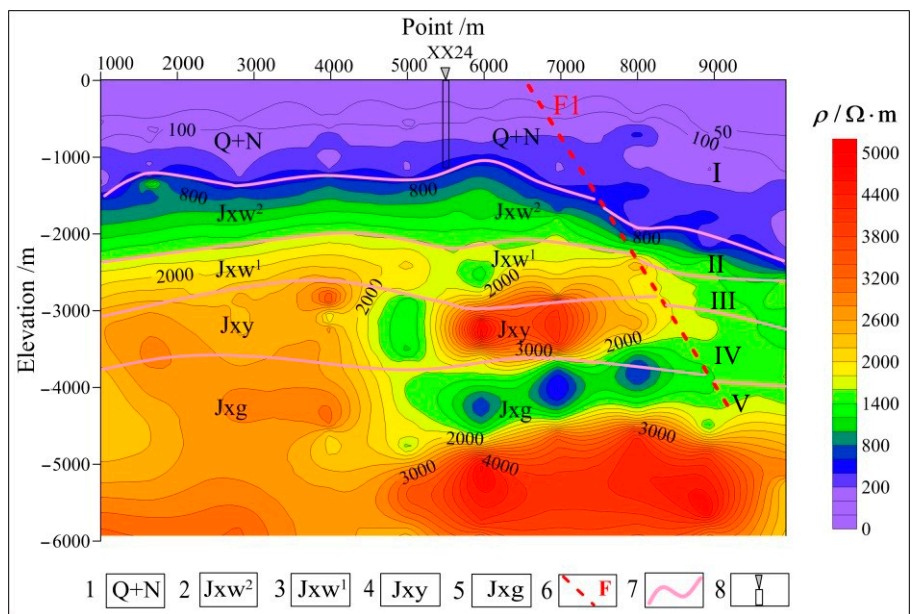

**Figure 3.** Integrated MT interpretation of the section of survey line L1. 1, Quaternary–Neogene; 2, upper component of the Jixianian Wumishan formation; 3, lower component of the Jixianian Wumishan formation; 4, Jixianian Yangzhuang formation; 5, Jixianian Gaoyuzhuang formation; 6, inferred fault; 7, stratigraphic boundary; 8, borehole.

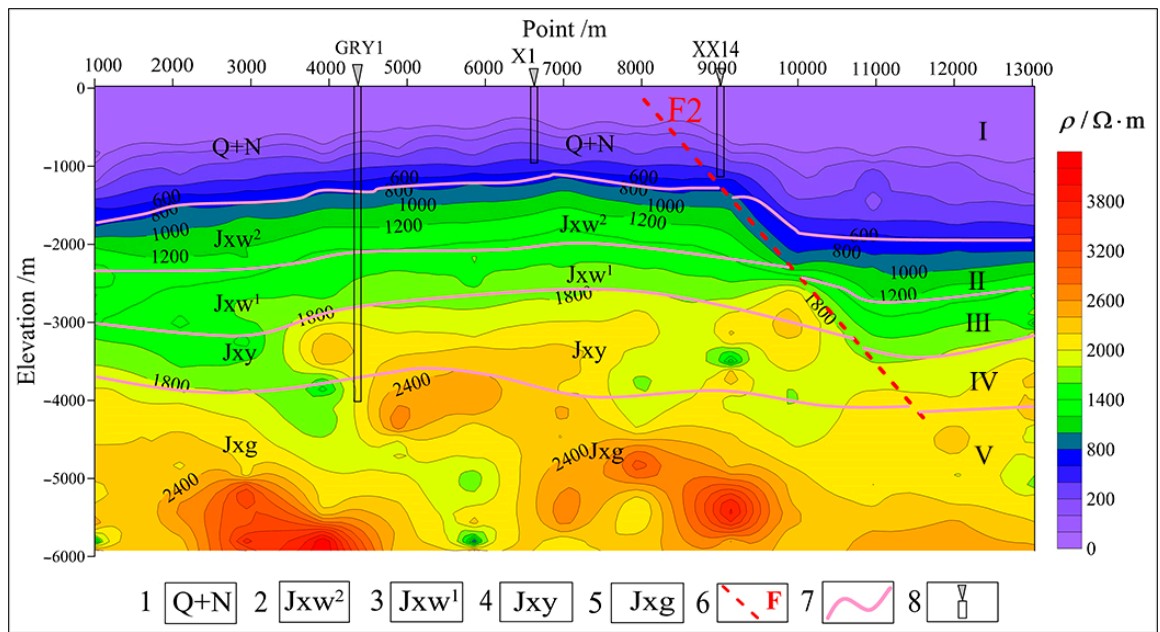

**Figure 4.** Integrated MT interpretation of a section of survey line L2 (refer to Figure 3 for an explanation of the symbols).

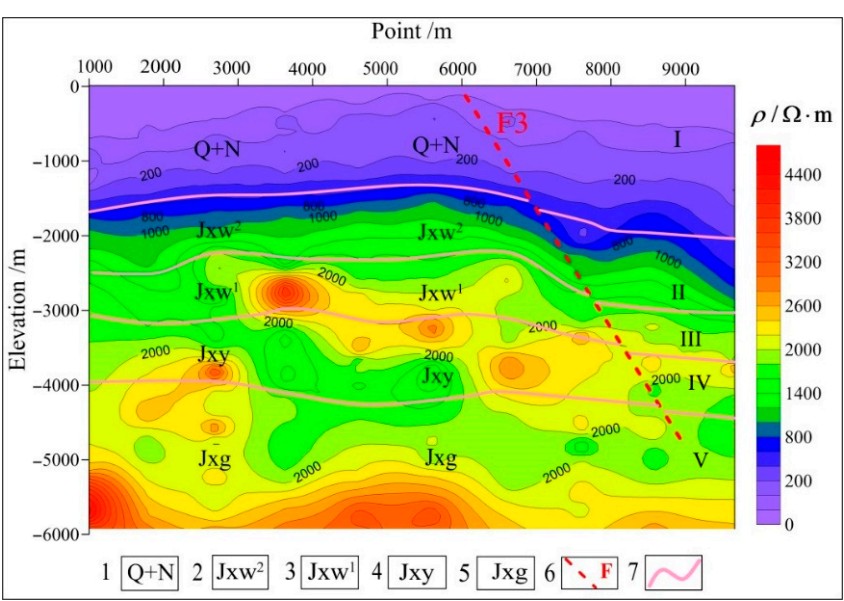

**Figure 5.** Integrated MT interpretation of a section of survey line L3 (refer to Figure 3 for an explanation of the symbols).

The integrated MT interpretation in this study was carried out by fully combining the regional geological plan and the data on the strata encountered during the drilling of the existing geothermal wells in the study area. As shown in Figure 3, the integrated MT interpretation of a section of survey line L1 showed a layered structure overall. The Cenozoic strata appear as light purple and blue zones in the figure, with a resistivity of under 800 $\Omega$·m. According to this result as well as the stratigraphic information revealed by borehole XX24 between measuring points at depths of 5000–6000 m, the Cenozoic strata have a burial depth of 1100–1500 m. The underlying light green and yellow areas have resistivities of 800–2500 $\Omega$·m. They reflect the upper and lower components of the Jixianian Wumishan formation and the Jixianian Yangzhuang formation. The borehole data show that (1) the upper component of the Wumishan formation mainly consists of light gray to grayish-white calcareous dolomites and has fissure veins, damaged cores, and a low resistivity of 800–1600 $\Omega$·m; (2) the lower component of the Wumishan formation mainly consists of interbeds consisting of dark gray dolomites and silty dolomites, with interbeds sandwiched with thick oolitic siliceous dolomites (this component has high resistivities of 1600–2500 $\Omega$·m due to tight rocks and a thickness of 700–1700 m, constituting one of the main geothermal reservoirs in the study area); (3) the Yangzhuang formation mainly consists of dark gray dolomites interbedded with purplish-red micritic dolomites and is characterized by the distribution of alternating high and low resistivities. The deep layers generally have high resistivities of 2500–4000 $\Omega$·m, which reflect the light gray to grayish-white siliceous dolomites that are rich in the banded chert of the Jixianian Gaoyuzhuang formation, with local intercalations of alternating black and white lamellar asphaltene dolomites. The horizon at depths of 4000–5000 m corresponding to surface point numbers from 5500 to 8000 shows local elliptical low-resistivity anomalies. On the basis of this result, as well as the drilling data, it can be inferred that the dolomites at this horizon have a high clay content, leading to a significant difference in the resistivity between the dolomites and the surrounding rocks. In addition, the resistivity contours bend at the horizon, with depths of 0–4500 m corresponding to the surface point numbers between 6500 and 8000, which were understood to correspond to fault F.

As shown in Figure 4, the integrated MT interpretation of the section of survey line L2 also shows a layered structure overall. Based on this and the known drilling data, the shallow portion of the section has low resistivities overall and reflects the Cenozoic strata. The middle part of the section has a high resistivity of 800–2400 $\Omega$·m and corresponds to the upper and lower components of the Jixianian Wumishan formation and the Jixianian

Yangzhuang formation. The deep portion shows high resistivity and is understood to reflect light gray to grayish-white siliceous dolomites of the Jixianian Gaoyuzhuang formation. The resistivity contours bend severely at the horizon at a depth of 0–4500 m, corresponding to the surface point numbers between 8000 and 10,000, which are likely due to fault F.

As shown in Figure 5, the integrated MT interpretation of the section of survey line L3 has obvious high and low resistivity zones and covers five strata longitudinally. It is inferred that the shallow low-resistivity zone is the Quaternary and Neogene strata. The middle portion corresponds to three strata, namely the upper and lower components of the Jixianian Wumishan formation and the Jixianian Yangzhuang formation. The upper and lower components of the Jixianian Wumishan formation consist of calcareous dolomites and silty dolomites, respectively. The increased rock density has resulted in a gradual increase in resistivity. The Jixianian Yangzhuang formation mainly includes interbeds consisting of grayish-white and thickly laminated dolomites, and purple silty micritic dolomites with a high clay content, leading to the uneven distribution and strong heterogeneity of resistivity. The deep part of this section shows high resistivity and might consist of the light gray to grayish-white dolomites of the Jixianian Gaoyuzhuang formation. The resistivity contours at depth of 0–5000 m in the zone corresponding to the surface point numbers between 6000 and 8000 m have a high density and a large inclination, which are likely due to fault F.

### 4.2. Estimation of the Geothermal Reservoir's Temperature

4.2.1. Hydrogeochemical Characteristics of Geothermal Waters

Table 2 shows the results of the hydrochemical analysis of geothermal water collected from wells XXZK1 and XXZK2 in this study. The geothermal waters from wells XXZK1 and XXZK2 had a pH of 7.1 and 7.2, respectively, and a TDS content of 5635 mg/L and 5910 mg/L, respectively, indicating moderately mineralized water. The cations and anions in the geothermal water were dominated by $Na^+$ and $Cl^-$, respectively; therefore, the waters have a hydrochemical type of Na-Cl. The TDS content of geothermal water is closely related to the state of its water–rock reactions. When geothermal water reaches an equilibrium state, the rocks show a high leaching intensity, and the TDS content of the geothermal water increases accordingly. The Na-K-Mg ternary diagram is commonly used to determine the state of water–rock interactions. As shown in Figure 6 (the Na-K-Mg ternary diagram of geothermal water in Xianxian County), the geothermal waters from wells XXZK1 and XXZK2 fell at the boundary between partially equilibrated water and immature water; hence, they have not reached an equilibrium state. In this case, the temperature of deep geothermal reservoirs evaluated using cation geothermometers will significantly deviate from the actual value.

**Table 2.** Hydrochemical analysis of geothermal waters from wells XXZK1 and XXZK2.

| Analysis Item $\rho\left(B^{z\pm}\right)/(\text{mg/L})$ | XXZK1 mg/L | XXZK2 mg/L | Analysis Item $\rho\left(B^{z\pm}\right)/(\text{mg/L})$ | XXZK1 mg/L | XXZK2 mg/L |
|---|---|---|---|---|---|
| $Na^+$ | 1798 | 1692 | $H_2SiO_3$ | 69.02 | 74.79 |
| $K^+$ | 90.19 | 93.46 | Free $CO_2$ | 30.61 | 23.99 |
| $Mg^{2+}$ | 38.83 | 42.91 | $P^H$ | 7.16 | 7.25 |
| $Ca^{2+}$ | 183.1 | 196.5 | Mn | 0.064 | 0.104 |
| $Fe^{2+}$ | - | - | Cu | <0.010 | <0.006 |
| $Cl^-$ | 2850 | 2582 | Total Cr | <0.020 | <0.020 |
| $SO_4^{2-}$ | 500.9 | 537.9 | Ba | 0.150 | 0.116 |
| $HCO_3^-$ | 347.8 | 353.4 | Li | 3.090 | 2.922 |
| $CO_3^{2-}$ | 0.00 | 0.00 | Ni | <0.008 | <0.008 |
| $F^-$ | 4.59 | 4.43 | Mo | 0.016 | 0.006 |
| $NO_3^-$ | <0.20 | 35.56 | TDS | 5910 | 5635 |

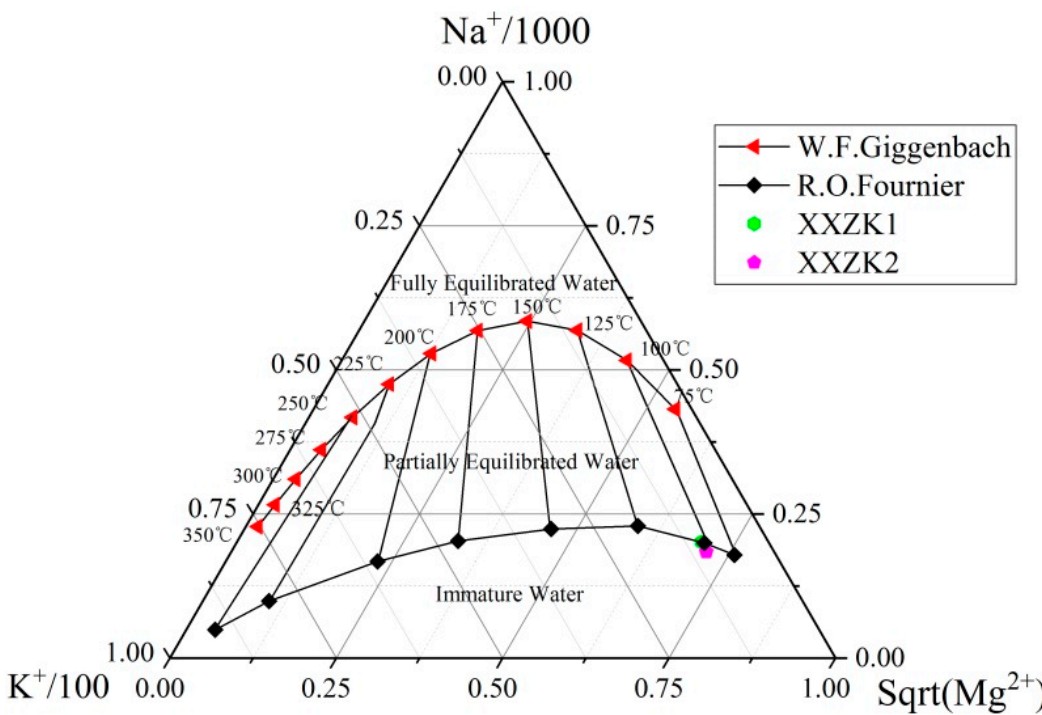

**Figure 6.** Na-K-Mg ternary diagram of geothermal waters in Xianxian County.

4.2.2. Resistivity-Based Inversion of the Geothermal Reservoir's Temperature

The resistivity of geothermal reservoirs is mainly related to factors such as the formation's lithology, the formation's porosity, permeability, the TDS content of groundwater, and the formation's temperature. Since temperature determines the TDS content of groundwater, the formation's temperature is an important parameter affecting the formation's resistivity. With an increase in the temperature, the density and viscosity of groundwater decrease, while the solubility of groundwater increases. Accordingly, the TDS content and ionic activity of groundwater increase, while the resistivity decreases. Figure 7 shows the relationships among the resistivity of water-bearing rock samples, temperature, and the TDS content of aqueous solutions [29]. As shown in Figure 7a, as the TDS content of the aqueous solution increases under the same temperature and pressure conditions, the formation's resistivity decreases to a certain value and then remains stable. As shown in Figure 7b, as the temperature increases under the same TDS content in the aqueous solution, the formation's resistivity also decreases. The Na-K-Mg ternary diagram (Figure 6) and Table 2 indicate that the geothermal water in Xianxian County has not reached the equilibrium state, the rocks show a low leaching intensity, and the groundwater has a low TDS content. Since the TDS content of the geothermal water is in the range where the formation's resistivity is sensitive to temperature changes, the formation's resistivity is significantly affected by changes in the temperature.

To conduct the resistivity-based inversion of temperature, it is necessary to understand the lithostratigraphy of the sections and the measured resistivity of the study area. As revealed by the integrated MT interpretation of sections of the survey lines L1, L2, and L3, the Xianxian strata include the Cenozoic (i.e., the Quaternary and Neogene) strata, the upper and lower components of the Jixianian Wumishan formation, the Jixianian Yangzhuang formation, micritic dolomites, and the Gaoyuzhuang formation from young to old overall. Moreover, the Xianxian strata can be divided into five zones (I–V) according to their lithology. Specifically, Zone I consists of sandstones, Zone II consists of light gray to grayish-white calcareous dolomites, Zone III consists of dark gray dolomites and silty dolomites, Zone IV consists of dark gray dolomites interbedded with purplish-red micritic dolomites, and Zone V consists of light gray to grayish-white dolomites containing chert nodules (Figures 3–5).

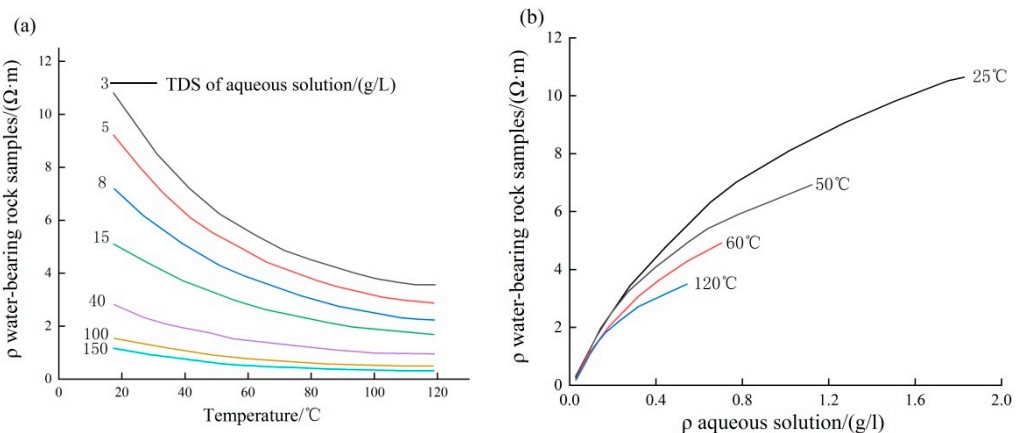

**Figure 7.** Relationships among the resistivity of water-bearing rock samples (**a**), temperature, and the TDS content of aqueous solutions (**b**).

According to the statistical table of the resistivity of formations in northern China (Table 1) and the existing data on rocks' resistivity, as well as the known borehole temperatures, the temperature coefficient of a rock (*a*) can be taken as 0.02. In practical applications, the average resistivity of a zone is generally set as its initial resistivity. To eliminate the influence of variations in the depth, this study calculated the initial resistivity at different depths using the linear interpolation method and appropriately adjusted the calculated resistivity based on the known resistivity of rocks and borehole temperatures. Finally, this study obtained the corresponding temperatures through an inversion of the resistivity based on the magnetotelluric data of survey lines L1, L2, and L3, as well as Equation (8), and prepared the temperature contour profiles using Surfer software (Figures 8–10).

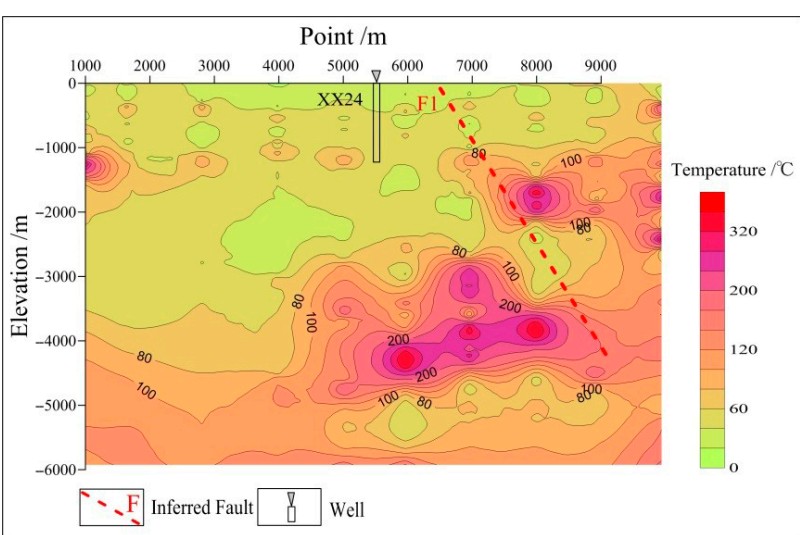

**Figure 8.** Temperature profile of survey line L1.

As shown in Figure 8, the temperature profile of survey line L1 shows medium and high temperatures overall and high-temperature anomalies along the faults. The areas corresponding to the surface point numbers from 6000 to 8000 have high temperatures of 200–240 °C at depths between 4000 and 5000 m, which are substantially different from the temperature at the bottom of borehole GRY1 (110 °C). Based on the integrated MT interpretation of the section of survey line A and the drilling data, the large temperature difference may be caused by dolomites with a high clay content. The portion at a depth of 2000–4000 m corresponding to the surface point numbers from 7000 to 10,000 also has high temperatures, which may be a result of the low-resistivity damage zone of fault

F1. The horizon at a depth of 4000 m has an average temperature of 80–110 °C, which is slightly different from the temperature at the bottom of borehole GRY1 and can be used as a scientific basis for estimating the temperature of the geothermal reservoirs in Xianxian County.

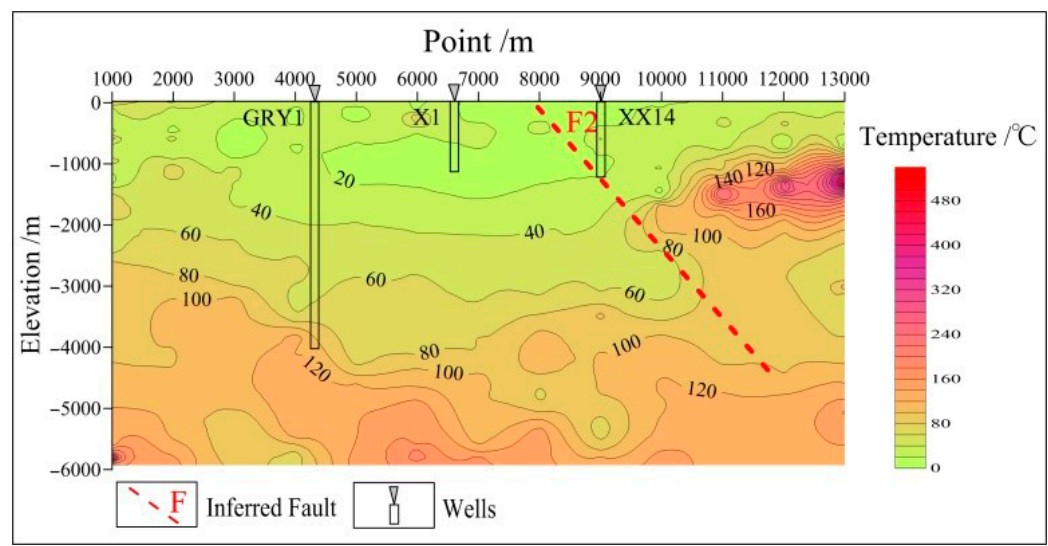

**Figure 9.** Temperature profile of survey line L2.

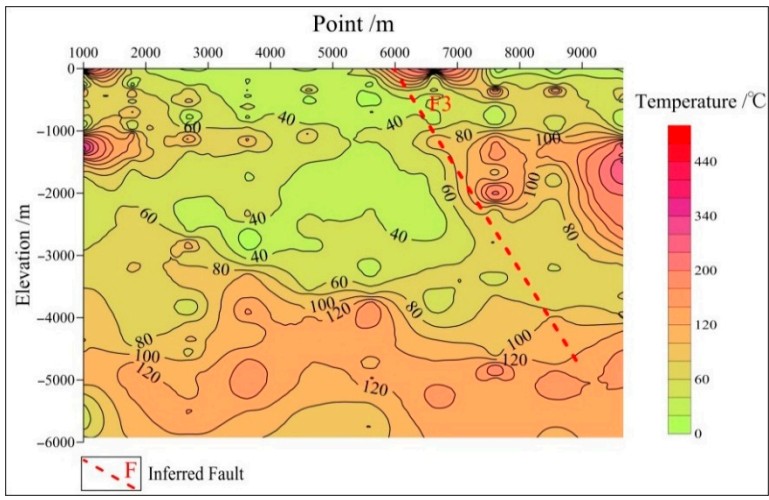

**Figure 10.** Temperature profile of survey line L3.

As shown in Figure 9, the temperature profile of survey line L2 reflects the presence of geothermal reservoirs and shows high temperatures in local areas near the faults. The temperature increases longitudinally in the temperature profile, which is consistent with the law that a formation's temperature gradually increases. The portion at a depth of 4000 m corresponding to the surface point numbers from 10,000 to 13,000 has a temperature of approximately 110 °C, which is consistent with the temperature at the bottom of borehole GRY1. Point No. 11,000 corresponds to higher temperatures of 120–180 °C, which is likely caused by the low-resistivity damage zone of fault F2. The portion at depths of 5000–6000 m has high temperatures of 120–140 °C, which can be used as a reference for estimating the temperature of deep geothermal reservoirs.

As shown in Figure 10, the temperature anomalies in the temperature profile of survey line L3 reflect the presence of geothermal reservoirs and high temperatures near the faults. The temperature profile shows low and high temperatures overall in the shallow and deep portions, respectively. The horizon at a depth of 4000 m corresponding to the surface point

numbers between 1000 and 9000 shows a temperature of approximately 100 °C, which is slightly different from the temperature of borehole GRY1 at the same depth (110 °C) and is consistent with the temperature response generated by the aquifers consisting of dolomites. The portion at a depth of 5000 m has temperatures of approximately 120–130 °C, which can be used as the temperature of the deep geothermal reservoirs. Point number 8000 corresponds to high temperatures at this depth, which may be caused by rock damage caused by fault F3 and the downward infiltration of surface water.

### 4.2.3. Calculation of Geothermal Reservoirs Using Geothermometers

(1)    Geochemical geothermometry

At present, the commonly used geothermometers include Na-K geothermometers, K-Mg geothermometers, and $SiO_2$ geothermometers, all of which apply to specific conditions. For instance, the geothermal water in the Xianxian geothermal field is partially equilibrated water. Moreover, the water–rock equilibrium of the geothermal water corresponds to a low temperature, and the geothermal water may be mixed with cold water. Therefore, it is inappropriate to estimate the geothermal reservoir's temperature using a Na-K geothermometer (Figure 6). By contrast, the K-Mg geothermometer is applicable to low-temperature geothermal water and can reach equilibrium in a solution more quickly than the Na-K geothermometer. Therefore, the K-Mg geothermometer can be used as a reference for determining the temperature of geothermal reservoirs in the Xianxian geothermal field. In addition, the precipitation rate of $SiO_2$ decreases with a decrease in the geothermal fluid's temperature. Therefore, the $SiO_2$ content can still indicate the underground concentration even if the geothermal fluids cool due to conduction-induced thermal loss. $SiO_2$ geothermometers can be divided into quartz geothermometers and chalcedony geothermometers. However, attention should be paid to the minerals controlling the $SiO_2$ concentration in a solution when using $SiO_2$ geothermometers. Arnórsson pointed out that the minerals controlling the concentration of $SiO_2$ are quartz at temperatures over 180 °C [40], chalcedony at temperatures below 110 °C, and both at temperatures between 110 and 180 °C. According to Table 2, the geothermal waters at the locations of wells XXZK1 and XXZK2 have an $H_2SiO_3$ concentration of 69.02 mg/L and 74.79 mg/L, respectively, from which the $SiO_2$ concentrations can be inferred to be 53 mg/L and 57 mg/L, respectively, according to Equation (11). On the basis of these results, the temperatures of the geothermal reservoir can be estimated, as shown in Table 3. The temperatures of the geothermal reservoir at the locations of wells XXZK1 and XXZK2 were estimated to be 75 °C and 79 °C, respectively, using chalcedony geothermometers, which are quite different from the temperature at the bottom of borehole GRY1 (108 °C). The geothermal water samples from the study area generally had lower temperatures than the local boiling point, and the temperatures of the geothermal reservoir at the locations of wells XXZK1 and XXZK2 were estimated to be 105 °C and 108 °C, respectively, using quartz geothermometers with no steam loss. These temperatures are consistent with the temperatures of the geothermal reservoir calculated using the hydrochemical data from borehole GRY1 [42]. The temperatures of the geothermal reservoir at the locations of geothermal wells XXZK1 and XXZK2 were calculated to be 104 °C for both with the K-Mg geothermometer, which is slightly different from the result obtained using the quartz geothermometer with no steam loss and thus can be used as a reference. Therefore, it is reasonable to use the temperatures of the geothermal reservoir (104–108 °C) estimated using the quartz geothermometer with no steam loss and the K-Mg geothermometer as the temperature of the deep geothermal reservoir in the study area. In addition, the temperature of the geothermal reservoir at a depth of 4000 m was estimated to be 100–110 °C using the resistivity–temperature method. This result is roughly consistent with the results estimated using the quartz geothermometer with no steam loss and the K-Mg geothermometer, indicating that the resistivity–temperature inversion method is feasible.

**Table 3.** Temperatures of the geothermal reservoir estimated using $SiO_2$ geothermometers and the K-Mg geothermometer.

| Geothermal Well | Temperature of the Geothermal Reservoir (°C) | | |
|---|---|---|---|
| | Chalcedony Geothermometer | Quartz Geothermometer with No-Steam Loss | K-Mg Geothermometer |
| XXZK1 | 75 | 105 | 104 |
| XXZK2 | 79 | 108 | 104 |

(2)  Multiple mineral equilibrium diagrams

By calculating the activity coefficients of water-soluble minerals in deep water on the basis of the water quality data of geothermal fluids, the PHREEQCI interface can simulate the formation of hydrochemical components and minerals in geothermal fluids, and also simulate the SI of minerals dissolved in the water. In the case of a fixed aluminum concentration [25], this study calculated the SI of various minerals at different temperatures and plotted the SI–T curves of geothermal wells XXZK1 and XXZK2 using PHREEQCI (Figure 11).

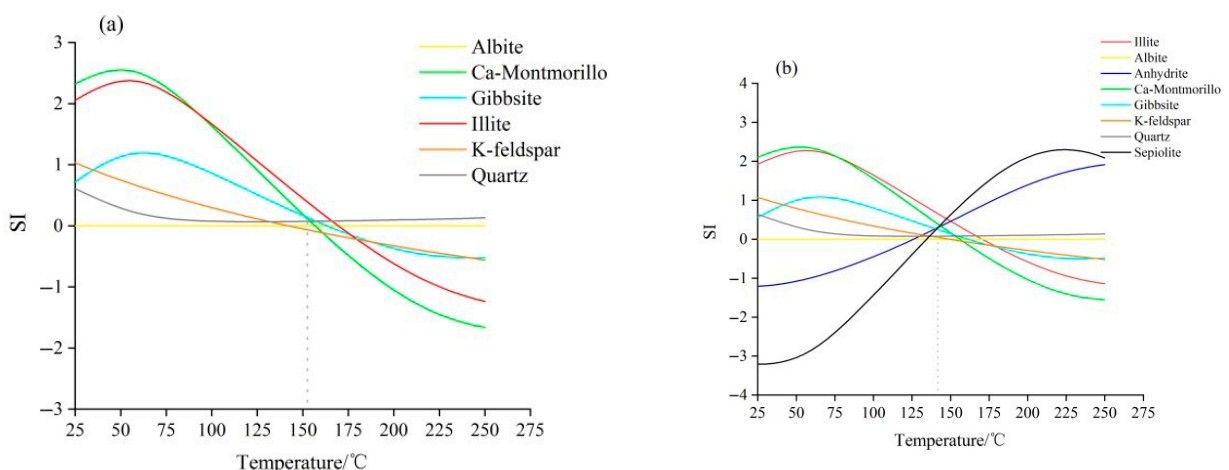

**Figure 11.** SI–T curves of various minerals from geothermal wells XXZK1 and XXZK2 in the Xianxian geothermal field: (**a**) XXZK1, (**b**) XXZK2.

According to Figure 11a, the temperature of the geothermal water in well XXZK1 is 140–160 °C under conditions of water–rock equilibrium, and the dominant minerals include albites, calcium montmorillonites, gibbsites, illites, K-feldspars, and quartz. This figure also shows that the temperature of the geothermal water in well XXZK2 is 135–160 °C under conditions of water–rock equilibrium, with the dominant minerals mainly including illites, albites, anhydrites, calcium montmorillonites, gibbsites, potassium feldspars, quartz, and sepiolites (Figure 11b). These results for wells XXZK1 and XXZK2 are roughly consistent with the temperatures at a depth of 5000–6000 m shown in the temperature sections of survey lines L1, L2, and L3 (Figures 8–10).

In summary, the temperatures of the geothermal reservoir calculated using different methods varied across wide ranges. This occurred because geothermal water is mixed with cold water during rising. The temperatures of the geothermal reservoir calculated using the quartz geothermometer with no steam loss, the K-Mg geothermometer, and the resistivity inversion method were the temperatures of deep geothermal reservoirs after mixing with cold water. By contrast, the temperatures of the geothermal reservoir obtained using the multiple mineral equilibrium diagrams may be the temperatures at a greater depth and were less influenced by the intrusion of cold water. Finally, the temperature

range of the deep geothermal reservoirs in the Xianxian geothermal field was determined to be 138–160 °C by averaging the temperatures of the geothermal reservoir estimated using the Multiple mineral equilibrium diagrams.

### 4.3. Circulation Depth of Geothermal Water

As one of the research topics of the circulation of geothermal water, the circulation depth of geothermal water reflects its recharging capacity. Specifically, a large circulation depth generally means that geothermal water resources have a strong recharging capacity and great potential for exploitation and utilization.

The geothermal waters in Xianxian County are generated as follows. The atmospheric precipitation infiltrates downward to recharge the groundwater; next, the recharged groundwater undergoes deep circulation and is heated by shallow geothermal energy to form geothermal water. In this case, the circulation depth of the geothermal water can be calculated using Equation (12). The study area has an average annual temperature of 13 °C. According to the temperature measurement curve of borehole GRY1 (Figure 12), the Xianxian geothermal field has a weighted average geothermal gradient of approximately 2.3 °C/100 m. Accordingly, the reciprocal of the geothermal gradient is 43 m/°C. The thickness of the constant temperature zone was taken to be 25 m. Substituting the average temperature of the geothermal reservoir (149 °C) into Equation (14) yielded a circulation depth of geothermal water in the Xianxian geothermal field of 5873 m.

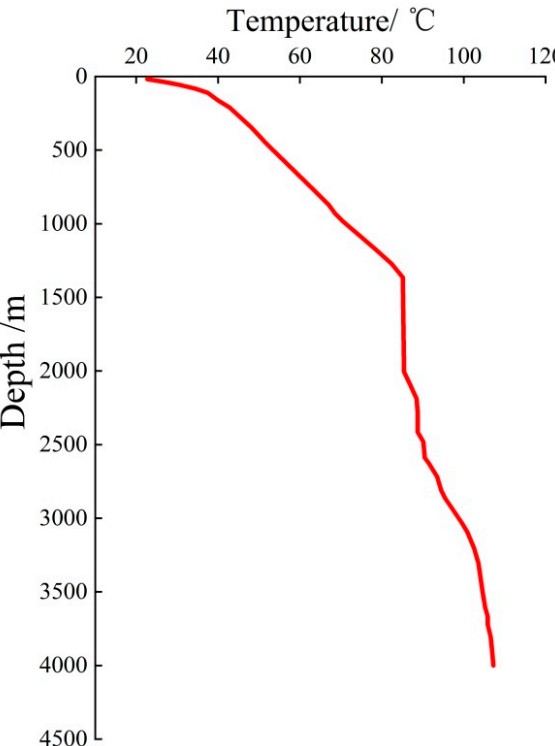

**Figure 12.** Temperature measurement curve of borehole GRY1.

## 5. Conclusions and Suggestions

### 5.1. Conclusions

(1) This comprehensive analysis based on magnetotelluric sounding and drilling data revealed that the geothermal reservoirs in the Xianxian geothermal field mainly include porous Neogene geothermal reservoirs and bedrock geothermal reservoirs with karst fissures. The latter are the most promising geothermal reservoirs in the study area and can be divided into the geothermal reservoirs of the Jixianian Wumishan and Gaoyuzhuang formations. The geothermal reservoirs of the Jixianian Wumishan formation have a top depth of 1100–1500 m and a thickness of 700–1700 m, and the ratio

of the thickness of the geothermal reservoirs to the formation is 15–30%. They mainly consist of dolomites and banded chert-rich dolomites. The geothermal reservoirs of the Jixianian Gaoyuzhuang formation have a top depth of 3700–4000 m, the thickness ratio of the geothermal reservoir to the formation is 15–30%, and the maximum drilled thickness is 400 m. They consist of dolomites and banded chert-rich dolomites.

(2)  This study established the MT resistivity–temperature relationship to obtain the field distribution of the temperature of geothermal reservoirs in the geothermal fields in sedimentary basins through inversion. The estimated field distribution of temperature is roughly consistent with the temperature measured in boreholes and the temperature estimated using the quartz geothermometer with no steam loss, the K-Mg geothermometer, and multiple mineral equilibrium diagrams, indicating that the resistivity–temperature inversion methodology is reliable. The geothermal reservoirs of the Xianxian geothermal field primarily have medium and low temperatures of 138–160 °C and the circulation depth of the geothermal water is 5873 m.

*5.2. Suggestions*

The geothermal fields in sedimentary basins have laterally continuous and consistent strata. Therefore, MT sounding can be used to effectively carry out the inversion of the stratigraphic distribution characteristics and infer the thickness of the geothermal reservoirs. The MT resistivity–temperature relationship established in this study is indicative of the temperature distribution of the geothermal reservoirs. The application of the MT method combined with fluid geochemistry allows for more effective inversion of the temperature and depth of the geothermal reservoirs and is significant for accurately evaluating the geothermal resources and improving the probability of penetration of geothermal reservoirs. This study explored the geothermal reservoirs of the Xianxian geothermal field in a sedimentary basin of northern China and achieved encouraging results. However, there are still some limitations, and we recommend that further research should be carried out in the following aspects.

(1)  To verify the accuracy of the temperature distribution of deep strata calculated using the MT resistivity–temperature inversion method, it is necessary to obtain more data on the lithology and resistivity of the strata through additional deep geothermal drilling. The purpose is to establish a more detailed and reliable formation resistivity-based inversion relationship and improve the accuracy and reliability of the resistivity–temperature inversion method.

(2)  The formation and evolution processes of the lithology and its geochemical features are critical to establishing the relationships among resistivity, fluid geochemistry, and the formation's temperature. In-depth research on these processes could shed light on the intrinsic relationships among the formation's lithology and resistivity, fluid geochemistry, and the geothermal reservoir's temperature. The purpose is to establish a more scientific, reliable, and comprehensive inversion method for deep geothermal reservoirs; explore the attributes of deep geothermal reservoirs from the surface more economically and effectively; and reduce the risks and costs of the exploration, development, and utilization of geothermal resources.

**Author Contributions:** Conceptualization, Y.L. and G.W.; methodology, J.H. and J.W.; software, G.Z.; formal analysis, X.G.; writing—original draft preparation, G.Z. and X.W.; writing—review and editing, Y.L., G.W., G.Z., J.H., X.G., X.W. and J.W. All authors have read and agreed to the published version of the manuscript.

**Funding:** This study was supported by the National Nature Science Foundation of China (No.4227020859), the National Key R&D Program of China (Grant Nos. 2021YFB1507402), the S&T Program of Hebei Province, China (No. 20374201D), the Foundation of the Institute of Hydrogeology and Environmental Geology of the Chinese Academy of Geological Sciences (No. SK202104), and the geological survey project of China (No. DD20221676-4).

**Institutional Review Board Statement:** Not required.

**Informed Consent Statement:** Not applicable.

**Data Availability Statement:** Not applicable.

**Conflicts of Interest:** The authors declare no conflict of interest.

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
