# Peer review of "A Joint Method Based on Geochemistry and Magnetotelluric Sounding for Exploring Geothermal Resources in Sedimentary Basins and Its Application"

_water, doi:10.3390/w14203299_

Round 1

Reviewer 1 Report

The manuscript presents the results of testing a new innovative method for exploration of geothermal resources. All sections of the manuscript contain the necessary explanations and justifications. When finalizing the manuscript, the following shortcomings should be eliminated. Remove paragraph (1) from the abstract. Bring figures to the same sizes frames and fonts. In the header of Table 2, replace the unit of ion concentrations with a more common one in hydrochemistry: mg/L (in plain type without italics).

Author Response

Thank you for the guidance. The manuscript has been carefully modified as follows according to your comments.

The manuscript presents the results of testing a new innovative method for exploration of geothermal resources. All sections of the manuscript contain the necessary explanations and justifications. When finalizing the manuscript, the following shortcomings should be eliminated. Remove paragraph (1) from the abstract. Bring figures to the same sizes frames and fonts. In the header of Table 2, replace the unit of ion concentrations with a more common one in hydrochemistry: mg/L (in plain type without italics).

Response: Tthe first sentence has been removed from the abstract and the unit format of ion concentrations has been modified.

Reviewer 2 Report

The manuscript introduces the internal connection between the resistivity and the temperature measured in geothermal wells, established a geothermal reservoir prospecting method based on multiple MT and geochemical information by combining geochemical thermometry.

The topic is interesting, and the paper has merit and would be worthy of publication after providing the following concerns:

1. Sedimentary basins, especially large ones, have favorable conditions for the occurrence of geothermal resources. The reserves of medium- and low-temperature geothermal resources account for more than 90% of the total geothermal resources in China. What are the advantages of joint method based on geochemistry and magnetotelluric sounding for exploring geothermal resources in sedimentary basins?

2. Introduction: The introduction was clearly presented and understandable. However, these articles reviewed was older. Try to show the novelty of your work.

3. How to quantitatively evaluate the reliability of MT resistivity inversion results? The residual curves should be included in your manuscript.

4. Supplement the relationship between Figures 3 and 4.

5. MT method is low to a middle-resolution method. The local low-resistivity anomaly in the inversion results is not considered to be a reliable underground structure anomaly. So it should be explained.

Author Response

Thank you for the guidance. The manuscript has been carefully modified as follows according to your comments.

The manuscript introduces the internal connection between the resistivity and the temperature measured in geothermal wells, established a geothermal reservoir prospecting method based on multiple MT and geochemical information by combining geochemical thermometry.

The topic is interesting, and the paper has merit and would be worthy of publication after providing the following concerns:

  1. Sedimentary basins, especially large ones, have favorable conditions for the occurrence of geothermal resources. The reserves of medium- and low-temperature geothermal resources account for more than 90% of the total geothermal resources in China. What are the advantages of joint method based on geochemistry and magnetotelluric sounding for exploring geothermal resources in sedimentary basins?

Response: The relationships of geothermal reservoir temperatures with the formation resistivity and geochemical reactions have been supplemented.

  1. Introduction: The introduction was clearly presented and understandable. However, these articles reviewed was older. Try to show the novelty of your work.

Response: Some latest references and relevant description have been supplemented.

  1. How to quantitatively evaluate the reliability of MT resistivity inversion results? The residual curves should be included in your manuscript.

Response: The MT resistivity inversion results were verified using temperatures measured at a depth of 4,000 m at a borehole.

  1. Supplement the relationship between Figures 3 and 4.

Response: Figures 3 and 4 show the MT integrated interpretation sections of two parallel survey lines, which are used for the comparative analysis of the structure of deep strata in the sedimentary basin.

  1. MT method is low to a middle-resolution method. The local low-resistivity anomaly in the inversion results is not considered to be a reliable underground structure anomaly. So it should be

Response: Local low-resistivity anomaly has been explained and supplemented in the manuscript.

Reviewer 3 Report

Dear Authors, dear Editor,

The manuscript entitled "A Joint Method Based on Geochemistry and Magnetotelluric Sounding for Exploring Geothermal Resources in Sedimentary Basins and Its Application" report new magnetotelluric and chemical data collected from two geothermal wells located at Xianxian geothermal field, China. The reported data represent an interesting contribution to the geothermal community and readers of Water. The manuscript is well written, but some grammar and word choice edits are needed.

The manuscript needs to be carefully revised. I suggest considering this manuscript for publication with major revisions after the authors have addressed the remarks and suggestions reported below.

 About content:

Introduction. In my opinion, the introduction section needs to include essential and recent scientific references related to your subject (i.e., MT methods used in geothermal exploration, application of geothermometers to evaluate the mineral equilibria, previous studies, etc.). Three references are reported for MT methods only. The introduction needs to include the scientific purpose of your study, how previous scientific studies have applied to achieve such goals, and whether there are limitations.

 Methods. I cannot identify the geothermal well selection. The authors report six geothermal wells in the manuscript, but only two wells were sampled and evaluated. It is not clear the purpose of selecting two geothermal wells only.

The water sampling methodology is not presented. I cannot identify overall questions that should be answered which justify the chemical composition. Please specify the buffer used for the pH-meter calibration and the techniques used for analysis (was it using a multi-probe in situ? Or was the pH measured in the lab?). How did you recollect the water sample? Did you use 0.45 µm membranes or 0.25 µm? Did you acidify the sample for cation analysis? What kind of acid did you use? What was the analytic technique for H2SiO3 analysis? Was the alkalinity determined by titration in situ? What were the detection limits for chemical analysis? In Table 2, please specify if the quality for major components was estimated by computing the charge imbalance. All this information is significant for geothermometry studies.

 Estimation of geothermal reservoir temperature.

In my opinion, these sections need to be incorporated and somewhat rewritten:  (a) first state what type of geochemical calculations were performed like calculation of mineral saturation index, calculations of geothermometry temperatures, (b) state what programs and databases, if relevant you used for this calculations; (c) describe better the calculations as needed in chronological order.  The above are classical geothermal exploration techniques suggested and developed mostly in the 70-80s.  But water composition in geothermal systems is considered to be controlled by (1) water-rock equilibria in some cases, (2) processes occurring from depth to surface like boiling, cooling, and condensation. These processes sometimes result in re-equilibration of water-rock equilibria, (3) mixing between thermal and shallow non-thermal water etc.  So, if you like to use geothermometry, you must demonstrate that your water is not affected by such secondary processes or correct them.  In fact, when looking at the NaKMg Giggenbach diagram, it is clear that the waters are close to equilibrium with K-Mg reaction. Some solute-mineral reactions may be at equilibrium, whereas others may not, and the water may be affected by dilution/mixing with non-thermal water.  So, what was the geochemical modeling approach to deal with these secondary processes and possibly re-equilibration? Nowadays, geothermometric modeling is applied and complemented with multicomponent geothermometry and the saturation index method for selecting the best geothermometers. Perhaps the following references may help to authors in this section:

 Pang, Z.H., Reed, M., 1998. Theoretical chemical thermometry on geothermal waters: problems and methods. Geochem. Cosmochim. Acta 6, 1083–1091.

Spycher, N., Peiffer, L., Sonnenthal, E.L., Saldi, G., Reed, M.H., Kennedy, B.M., 2014. Integrated multicomponent solute geothermometry. Geothermics 51, 113–123.

Pandarinath, K., Dominguez-Dominguez, H. 2015. Evaluation of the solute geothermometry of thermal springs and drilled wells of La Primavera (Cerritos Colorados) geothermal field, Mexico: A geochemometrics approach. Journal of South American Earth Science 62, 109-124.

Jácome-Paz, M.P., Pérez-Zárate, D., Prol-Ledesma, R.M., González- Romo, I.A., Rodríguez-Díaz, A., 2022. Geochemical Exploration in Mesillas geothermal area, Nayarit, Mexico. . Appl. Geochem. 143, 105376

 Results

 Line: 383: I suggest describing the interpolation method in the methods section. What do you mean by "adjusted the calculated resistivity"?

 The SiO2 concentration by the authors is inconsistent with equation 11:

 Equation 11, well XXK1:  60 x (69.02 / 78) = 53.09 mg/l 56.09 mg/l    line 440

 Why did you select the SiO2 geothermometer only? It is not clear the purpose of selecting this solute geothermometer. It would be good to review the recent geothermometry developments. The K/Mg geothermometer results are also realistic: XXK1= XXK2 = 104 °C. This geothermometer applies to situations where dissolved Na and Ca are not equilibrated between fluid and rock. This geothermometer re-equilibrates quickly at cooler temperatures related to the equation 2.

 About spelling and grammar:

Concerning language, the spelling and grammar need to be corrected. I show some suggestions in the pdf file (yellow marks).

 The style of references is different in the text according to the Journal's instructions. I point it out in the pdf file (blue marks).

 The reported numbers in the manuscript are not consistent (i.e., 1000 or 1,000); the authors must be chosen one style only in the manuscript. I point it out in the pdf file as red marks.

 The symbol alpha appears as "a" in both equation 7 and equation 8.

 About Tables and Figures

 From Table 1, I cannot access reference 34: Zhang, J. MT Sounding and Geothermal Resource Assessment in XianXian Convex of Bohai Bay Basin. China University of 611 Geosciences(Beijing), 2014. So, I have two questions: (1) the average resistivity is a mean value?, and (2) If it is the mean value, why is the standard deviation not reported in Table 1?

 The manuscript has figures of high quality; however, the legend in Figure 2 has two different fonts and size styles. I suggest improving the legend in Figure 2.

 I suggest the following paragraph as part of the Figure 3 caption:

"1- Quaternary‒Neogene; 2- Upper member of the Jixianian Wumishan Formation; 3- Lower member of the Jixianian Wumishan Formation; 4- Jixianian Yangzhuang Formation; 5- Jixianian Gaoyuzhuang Formation; 6- Inferred fault; 7- Stratigraphic boundary; 8- Borehole"

The following paragraph (lines 278 to 280) is repeated. I suggest writing it as part of Figure 4 caption:

"1- Quaternary‒Neogene; 2- Upper member of the Jixianian Wumishan Formation; 3- Lower member of the Jixianian Wumishan Formation; 4- Jixianian Yangzhuang Formation; 5- Jixianian Gaoyuzhuang Formation; 6- Inferred fault; 7- Stratigraphic boundary; 8- Borehole"

The following paragraph (lines 293 to 295) is also repeated. I suggest writing it as part of Figure 5 caption or making the reference to Figure 3:

"1- Quaternary‒Neogene; 2- Upper member of the Jixianian Wumishan Formation; 3- Lower member of the Jixianian Wumishan Formation; 4- Jixianian Yangzhuang Formation; 5- Jixianian Gaoyuzhuang Formation; 6- Inferred fault; 7- Stratigraphic boundary"

Figure 3 and Figure 8 can be joined into one figure, as well as Figure 4 with Figure 9 and Figure 5 with Figure 10.

In summary, the authors are suggested to justify the SiO2 geothermometer selection. Because traditional geochemistry approaches are employed to explore a traditional geothermal field, a comparison with newly improved solute geothermometers and geothermometric modeling would be helpful to increase the novelty and significance of this study.

Author Response

Thank you for the guidance. The manuscript has been carefully modified one by one as follows according to your comments.

The manuscript entitled "A Joint Method Based on Geochemistry and Magnetotelluric Sounding for Exploring Geothermal Resources in Sedimentary Basins and Its Application" report new magnetotelluric and chemical data collected from two geothermal wells located at Xianxian geothermal field, China. The reported data represent an interesting contribution to the geothermal community and readers of Water. The manuscript is well written, but some grammar and word choice edits are needed.

The manuscript needs to be carefully revised. I suggest considering this manuscript for publication with major revisions after the authors have addressed the remarks and suggestions reported below.

 About content:

Introduction. In my opinion, the introduction section needs to include essential and recent scientific references related to your subject (i.e., MT methods used in geothermal exploration, application of geothermometers to evaluate the mineral equilibria, previous studies, etc.). Three references are reported for MT methods only. The introduction needs to include the scientific purpose of your study, how previous scientific studies have applied to achieve such goals, and whether there are limitations.

Response: The applications of the MT method in the geothermal exploration have been further summarized, the shortcomings of the method in these applications have been supplemented, and the relevant references have also been supplemented.

 Methods. I cannot identify the geothermal well selection. The authors report six geothermal wells in the manuscript, but only two wells were sampled and evaluated. It is not clear the purpose of selecting two geothermal wells only.

The water sampling methodology is not presented. I cannot identify overall questions that should be answered which justify the chemical composition. Please specify the buffer used for the pH-meter calibration and the techniques used for analysis (was it using a multi-probe in situ? Or was the pH measured in the lab?). How did you recollect the water sample? Did you use 0.45 µm membranes or 0.25 µm? Did you acidify the sample for cation analysis? What kind of acid did you use? What was the analytic technique for H2SiO3 analysis? Was the alkalinity determined by titration in situ? What were the detection limits for chemical analysis? In Table 2, please specify if the quality for major components was estimated by computing the charge imbalance. All this information is significant for geothermometry studies.

Response: The sampling and test methods of hydrochemical samples have been supplemented.

 Estimation of geothermal reservoir temperature.

In my opinion, these sections need to be incorporated and somewhat rewritten:  (a) first state what type of geochemical calculations were performed like calculation of mineral saturation index, calculations of geothermometry temperatures, (b) state what programs and databases, if relevant you used for this calculations; (c) describe better the calculations as needed in chronological order.  The above are classical geothermal exploration techniques suggested and developed mostly in the 70-80s.  But water composition in geothermal systems is considered to be controlled by (1) water-rock equilibria in some cases, (2) processes occurring from depth to surface like boiling, cooling, and condensation. These processes sometimes result in re-equilibration of water-rock equilibria, (3) mixing between thermal and shallow non-thermal water etc.  So, if you like to use geothermometry, you must demonstrate that your water is not affected by such secondary processes or correct them.  In fact, when looking at the NaKMg Giggenbach diagram, it is clear that the waters are close to equilibrium with K-Mg reaction. Some solute-mineral reactions may be at equilibrium, whereas others may not, and the water may be affected by dilution/mixing with non-thermal water.  So, what was the geochemical modeling approach to deal with these secondary processes and possibly re-equilibration? Nowadays, geothermometric modeling is applied and complemented with multicomponent geothermometry and the saturation index method for selecting the best geothermometers. Perhaps the following references may help to authors in this section:

 Pang, Z.H., Reed, M., 1998. Theoretical chemical geothermometry on geothermal waters: problems and methods. Geochem. Cosmochim. Acta 6, 1083–1091.

Spycher, N., Peiffer, L., Sonnenthal, E.L., Saldi, G., Reed, M.H., Kennedy, B.M., 2014. Integrated multicomponent solute geothermometry. Geothermics 51, 113–123. doi.org/10.1016/j.geothermics.2013.10.012

Pandarinath, K., Dominguez-Dominguez, H. 2015. Evaluation of the solute geothermometry of thermal springs and drilled wells of La Primavera (Cerritos Colorados) geothermal field, Mexico: A geochemometrics approach. Journal of South American Earth Science 62, 109-124, 10.1016/S0016-7037(98)00037-4

Jácome-Paz, M.P.; Pérez-Zárate, D.; Prol-Ledesma, R.M.; González- Romo, I.A.; Rodríguez-Díaz, A., Geochemical Exploration in Mesillas geothermal area, Nayarit, Mexico. Appl. Geochem, 2022, 143, 105376, 10.1016/j.apgeochem.2022.105376

Response: The calculation method of the geothermal reservoir temperature has been supplemented.

 Results

 Line: 383: I suggest describing the interpolation method in the methods section. What do you mean by "adjusted the calculated resistivity"?

 The SiO2 concentration by the authors is inconsistent with equation 11:

 Equation 11, well XXK1:  60 x (69.02 / 78) = 53.09 mg/l ≠ 56.09 mg/l    line 440

 Why did you select the SiO2 geothermometer only? It is not clear the purpose of selecting this solute geothermometer. It would be good to review the recent geothermometry developments. The K/Mg geothermometer results are also realistic: XXK1= XXK2 = 104 °C. This geothermometer applies to situations where dissolved Na and Ca are not equilibrated between fluid and rock. This geothermometer re-equilibrates quickly at cooler temperatures related to the equation 2.

Response: Thanks for your suggestions, and the manuscript has been modified accordingly. The condition that the geothermal water is mixed with cold water during rising has been analyzed using the K/Mg geothermometer and the integrated multicomponent solute geothermometry.

 About spelling and grammar:

Concerning language, the spelling and grammar need to be corrected. I show some suggestions in the pdf file (yellow marks).

 The style of references is different in the text according to the Journal's instructions. I point it out in the pdf file (blue marks).

 The reported numbers in the manuscript are not consistent (i.e., 1000 or 1,000); the authors must be chosen one style only in the manuscript. I point it out in the pdf file as red marks.

 The symbol alpha appears as "a" in both equation 7 and equation 8.

Response: The manuscript has been modified carefully according to this comment.

 About Tables and Figures

 From Table 1, I cannot access reference 34: Zhang, J. MT Sounding and Geothermal Resource Assessment in XianXian Convex of Bohai Bay Basin. China University of 611 Geosciences(Beijing), 2014. So, I have two questions: (1) the average resistivity is a mean value?, and (2) If it is the mean value, why is the standard deviation not reported in Table 1?

 The manuscript has figures of high quality; however, the legend in Figure 2 has two different fonts and size styles. I suggest improving the legend in Figure 2.

 I suggest the following paragraph as part of the Figure 3 caption:

"1- Quaternary‒Neogene; 2- Upper member of the Jixianian Wumishan Formation; 3- Lower member of the Jixianian Wumishan Formation; 4- Jixianian Yangzhuang Formation; 5- Jixianian Gaoyuzhuang Formation; 6- Inferred fault; 7- Stratigraphic boundary; 8- Borehole"

The following paragraph (lines 278 to 280) is repeated. I suggest writing it as part of Figure 4 caption:

"1- Quaternary‒Neogene; 2- Upper member of the Jixianian Wumishan Formation; 3- Lower member of the Jixianian Wumishan Formation; 4- Jixianian Yangzhuang Formation; 5- Jixianian Gaoyuzhuang Formation; 6- Inferred fault; 7- Stratigraphic boundary; 8- Borehole"

The following paragraph (lines 293 to 295) is also repeated. I suggest writing it as part of Figure 5 caption or making the reference to Figure 3:

"1- Quaternary‒Neogene; 2- Upper member of the Jixianian Wumishan Formation; 3- Lower member of the Jixianian Wumishan Formation; 4- Jixianian Yangzhuang Formation; 5- Jixianian Gaoyuzhuang Formation; 6- Inferred fault; 7- Stratigraphic boundary"

Figure 3 and Figure 8 can be joined into one figure, as well as Figure 4 with Figure 9 and Figure 5 with Figure 10.

In summary, the authors are suggested to justify the SiO2 geothermometer selection. Because traditional geochemistry approaches are employed to explore a traditional geothermal field, a comparison with newly improved solute geothermometers and geothermometric modeling would be helpful to increase the novelty and significance of this study.

Response: The tables and figures have been modified carefully according to this comment. The geothermometers have been justified.

Round 2

Reviewer 3 Report

Dear Authors, Dear Editor:

I have completed the manuscript revision. I consider the manuscript has improved: the methodology is correctly described, and the equilibrium temperature estimates using geothermometry methods support the presented results.

I appreciate the authors’ effort to address all the comments.

I consider that the article can be published.